# Functional, Metabolic and Morphologic Results of Ex Vivo Donor Lung Perfusion with a Perfluorocarbon-Based Oxygen Carrier Nanoemulsion in a Large Animal Transplantation Model

**DOI:** 10.3390/cells9112501

**Published:** 2020-11-18

**Authors:** Ilhan Inci, Stephan Arni, Ilker Iskender, Necati Citak, Josep Monné Rodriguez, Miriam Weisskopf, Isabelle Opitz, Walter Weder, Thomas Frauenfelder, Marie Pierre Krafft, Donat R. Spahn

**Affiliations:** 1Department of Thoracic Surgery, University Hospital Zurich–University of Zurich, CH-8091 Zurich, Switzerland; stephan.arni@usz.ch (S.A.); ilker.iskender@yahoo.ca (I.I.); necomomus@gmail.com (N.C.); Isabelle.Schmitt-Opitz@usz.ch (I.O.); w.weder@thorax-zuerich.ch (W.W.); 2Institute of Veterinary Pathology, Vetsuisse Faculty, University of Zurich, CH-8057 Zurich, Switzerland; josep.monnerodriguez@uzh.ch; 3Department of Surgical Research, University Hospital Zurich–University of Zurich, CH-8091 Zurich, Switzerland; miriam.weisskopf@usz.ch; 4Institute of Radiology, University Hospital Zurich–University of Zurich, CH-8091 Zurich, Switzerland; thomas.frauenfelder@usz.ch; 5Institute Charles Sadron, CNRS, University of Strasbourg, 67200 Strasbourg, France; marie-pierre.krafft@ics-cnrs.unistra.fr; 6Institute of Anesthesiology, University Hospital Zurich-University of Zurich, CH-8091 Zurich, Switzerland; donat.spahn@usz.ch

**Keywords:** ex vivo lung perfusion, oxygen carrier, perfluorocarbon, lung transplantation

## Abstract

Background: Ex vivo lung perfusion (EVLP) is a technology that allows the re-evaluation of questionable donor lung before implantation and it has the potential to repair injured donor lungs that are otherwise unsuitable for transplantation. We hypothesized that perfluorocarbon-based oxygen carrier, a novel reconditioning strategy instilled during EVLP would improve graft function. Methods: We utilized perfluorocarbon-based oxygen carrier (PFCOC) during EVLP to recondition and improve lung graft function in a pig model of EVLP and lung transplantation. Lungs were retrieved and stored for 24 h at 4 °C. EVLP was done for 6 h with or without PFCOC. In the transplantation groups, left lung transplantation was done after EVLP with or without PFCOC. Allograft function was assessed by means of pulmonary gas exchange, lung mechanics and vascular pressures, histology and transmission electron microscopy (TEM). Results: In the EVLP only groups, physiological and biochemical markers during the 6-h perfusion period were comparable. However, perfusate lactate potassium levels were lower and ATP levels were higher in the PFCOC group. Radiologic assessment revealed significantly more lung infiltrates in the controls than in the PFCOC group (*p* = 0.04). In transplantation groups, perfusate glucose consumption was higher in the control group. Lactate levels were significantly lower in the PFCOC group (*p* = 0.02). Perfusate flavin mononucleotide (FMN) was significantly higher in the controls (*p* = 0.008). Post-transplant gas exchange was significantly better during the 4-h reperfusion period in the PFCOC group (*p* = 0.01). Plasma IL-8 and IL-12 levels were significantly lower in the PFCOC group (*p* = 0.01, *p* = 0.03, respectively). ATP lung tissue levels at the end of the transplantation were higher and myeloperoxidase (MPO) levels in lung tissue were lower in the PFCOC group compared to the control group. In the PFCOC group, TEM showed better tissue preservation and cellular viability. Conclusion: PFCOC application is safe during EVLP in lungs preserved 24 h at 4 °C. Although this strategy did not significantly affect the EVLP physiology, metabolic markers of the donor quality such as lactate production, glucose consumption, neutrophil infiltration and preservation of mitochondrial function were better in the PFCOC group. Following transplantation, PFCOC resulted in better graft function and TEM showed better tissue preservation, cellular viability and improved gas transport.

## 1. Introduction

Due to graft shortage, approximately 10–13% of lung transplant candidates die while on the waiting list every year [1,2,3] Additionally, there is a low lung utilization rate of 15–20% from eligible multiorgan donors [4]. This utilization rate is lower than in other solid organs such as liver and kidney because the donor lung is very susceptible to injury [5] The use of marginal donors [6], lobar transplantation [7], living-related donors [8], donation after cardiac death (DCD) donors [9,10] and ex vivo lung perfusion [11] are alternatives to alleviate donor shortage in lung transplantation. 

Ex vivo lung perfusion (EVLP) has become an alternative to conventional approaches for donor lung assessment, reconditioning, and preservation in clinical lung transplantation [12].

EVLP is a technology that allows the re-evaluation of questionable donor lung before implantation and it has the potential to repair injured donor lungs that are otherwise not suitable for transplantation [13,14,15]. 

Normothermic ex vivo organ perfusion was reported as early as 1935 [16]. Steen et al. introduced the concept of ex vivo reperfusion as a method, not only to evaluate lungs before transplantation, but also to revitalize lungs of inferior quality outside the cadaver and to extend the cold ischemic time by intermittent warm reperfusion [17]. After extensive research in their lab, Steen et al. reported successful lung transplantation from uncontrolled DCD donors [18]. This case report encouraged many transplant research groups [19,20,21,22,23,24,25,26,27], including our lab in Zurich [28,29,30,31,32,33,34,35] to further investigate and improve the role of EVLP in the field of lung transplantation.

The goal of developing oxygen (O_2_) carrying blood substitutes has evolved from replicating blood O_2_ transports properties to that of preserving microvascular and organ function, reducing the inherent or potential toxicity of the material used to carry O_2_, and treating pathologies initiated by anemia and hypoxia. Furthermore, the emphasis has shifted from blood replacement fluid to so-called “O_2_ therapeutics” that restore tissue oxygenation to specific tissues regions [36,37]. Perfluorocarbons (PFCs) are chemically and biologically inert, and can dissolve large amounts of O_2_ and other gases [36,38,39] with no possibility of chemical binding and interference from CO, NO and other reagents. Moreover, as O_2_ is released, carbon dioxide (CO_2_) is taken up. Oxygen dissolved in PFC is immediately available to tissues with a high extraction ratio [36,37]. In addition, several studies have suggested that systemic administration of PFC emulsion has anti-inflammatory effects [40,41,42,43,44,45].

In this study, we hypothesized that perfluorocarbon-based oxygen carrier, a novel reconditioning strategy instilled during EVLP would improve graft function. We tested our hypothesis in a pig left lung transplantation model.

## 2. Materials and Methods

We utilized perfluorocarbon-based oxygen carrier (PFCOC) during EVLP to recondition and improve lung graft function in a large animal model of EVLP and lung transplantation.

All animals received humane care during the experiments in accordance with the updated “The Guide for the Care and Use of Laboratory Animals” (8th Edition, US National Research Council). The Canton of Zurich Veterinary Authorities approved the animal use protocol (ZH 150/16).

### 2.1. Study Groups

(1)EVLP Only Groups
(i)PFCOC: Lungs were retrieved and stored for 24 h at 4 °C. PFCOC emulsion was added to the EVLP circuit (n = 7). EVLP time was 6 h.(ii)Control: Lungs were retrieved and stored for 24 h at 4 °C. EVLP was performed without PFCOC (n = 6). EVLP time was 6 h.(2)Transplantation Groups
(i)Control: Lungs were retrieved and stored for 24 h at 4 °C. EVLP was performed without PFCOC. The left lung was transplanted after 6 h of EVLP. Observation of the transplanted lung was 4 h (n = 5).(ii)PFCOC: Lungs were retrieved and stored for 24 h at 4 °C. PFCOC emulsion was added to the EVLP. The left lung was transplanted after 6 h of EVLP. Observation of the transplanted lung was 4 h (n = 4).(iii)Immediate Transplantation (Im-Tx): Following 4 h of cold ischemic storage, the left lung was transplanted. Observation of the transplanted lung was 4 h (n = 3).

### 2.2. Lung Retrieval, EVLP, and Lung Transplantation Procedures

Domestic female pigs (25–35 kg) were used for all study phases. Pig lungs were randomly assigned into study groups. Donor lungs retrieved using standard protocol and preserved at 4 °C for 24 h with Perfadex (XVIVO Perfusion, Göteborg, Sweden) followed by EVLP according to a previous report [34]. Lungs were perfused in EVLP system for 6 h according to the Toronto protocol [11].

In transplantation groups, the recipient operation was started after 5 h of EVLP. After thoracotomy, the right pulmonary artery and right main bronchus were encircled followed by a left pneumonectomy. Orthotopic left single lung transplantation was then performed and the recipients were observed for 4 h.

### 2.3. Physiological Assessment During EVLP and after Transplantation

The allograft function was assessed hourly during the 6 h EVLP and after reperfusion by means of pulmonary gas exchange, lung mechanics and vascular pressures. In the transplantation groups, at the end of 4 h of reperfusion, the allograft was challenged with a contralateral pulmonary artery occlusion for 5 min, followed by clamping of the right main bronchus for evaluation of the isolated allograft mechanics.

### 2.4. Perfusate, Plasma, Bronchoalveolar Lavage (BAL), and Tissue Collection and Analyses

Perfusate and plasma samples were collected hourly. The Epoc blood analysis system (Epocal, Inc., Ottawa, ON, Canada) was used to measure perfusate and blood gases and for clinical ions biochemistry.

At the end of EVLP and transplantation, a bronchoalveolar lavage was performed with 20 mL of cold saline from the lower lobe segments of the right and left lungs, respectively. Approximately 15 mL were recovered during each wash. The recovered wash fluid was subjected to cytological assessment at the hospital’s laboratories, using May–Grünwald–Giemsa stained cytological specimens and were subsequently processed as previously described for further analysis [30]. Similarly, lung tissue samples were also collected at the end of each procedure.

### 2.5. Assessment of the Inflammatory Response

The inflammatory response in perfusate, plasma, and BAL samples were profiled using a 13-plex porcine cytokine/chemokine array (Porcine Cytokine Array/Chemokine Array 13-Plex Panel; Cat no: PD13, Eve Technologies, Alberta, Canada). 

### 2.6. Myeloperoxidase Activity Measurement

Lung tissues were collected and flash frozen in liquid nitrogen and then stored at −80 °C. Powdered frozen lung tissue (25 mg) were homogenized in 0.5 mL of 0.5% trichloroacetic acid and centrifuged at 8000 rpm for 2 min at 4 °C. The supernatant was isolated and 10× concentrated Tris-acetate buffer was used to neutralize pH to 7.4 with a 10 mL of 0.002% xylenol blue used as the pH indicator. Myeloperoxidase (MPO) activity was measured using a kit (BioVision Inc cat K744-100) according to the manufacturer’s instructions, and the results were expressed in milliunits per milliliter.

### 2.7. Estimate of Tissue ATP Content

We used an ATP assay kit (Enliten, Promega, Madison, WI, USA) to enzymatically estimate the ATP concentration in the supernatant by measuring it in the luminescence channel of a Cytation 5 plate reader (BioTek Instruments, Inc., Winooski, VT, USA). Lung tissues were collected and flash frozen in liquid nitrogen and then stored at −80 °C. Frozen lung tissues (25 mg) were homogenized in 0.5 mL of 0.5% trichloroacetic acid and centrifuged at 8000 rpm for 2 min at 4 °C. The supernatant was isolated and 10× concentrated Tris-acetate buffer was used to neutralize the pH to 7.4 with 10 mL of 0.002% xylenol blue used as the pH indicator. A microBCA protein assay kit was used with bovine serum albumin as standard to determine protein concentration in the supernatant of each sample tested according to the manufacturer’s instructions (Thermo Scientific, Rockford, IL, USA).

### 2.8. Flavin Mononucleotide Measurement

The microplate was analyzed by fluorescence spectroscopy. A monochrome light with an excitation wavelength of 485 nm was used and the fluorescence with a wavelength of 528 nm emitted by the FMN was recorded with 100% gain. The fluorescence readings in the artificial unit (AU) were displayed in an Excel spreadsheet produced by the spectrometer. This experiment was performed 3 times [46].

### 2.9. Radiologic Assessment of the Lungs

At the end of EVLP a radiologist evaluated the lung X-rays in a blinded fashion as previously described [33].

### 2.10. Histological Assessment of the Lungs

Tissue samples taken at the end of EVLP and transplantation were stained with hematoxylin and eosin and blindly assessed by two veterinary pathologists for any pathological changes. The most prominent features observed in the transplanted lungs were used to develop a scoring system which was as follows: (A) Evidence of neutrophil recruitment into the lungs, represented by the presence of neutrophils within vessels (1: moderate numbers, 2: high numbers, 3: vessels packed with neutrophils); (B) Evidence of neutrophil emigration into the tissue (1: neutrophil rolling along vascular endothelial cells, 2: neutrophils within vessel walls, 3: neutrophils immediately outside vessels); and (C) Evidence of neutrophils within the tissue, represented by the presence of individual neutrophils within alveolar lumen (1: in rare alveoli, 2: in occasional alveoli, 3: in numerous alveoli).

### 2.11. Transmission Electron Microscopy (TEM)

Transmission electron tissue samples were fixed in 2.5% glutaraldehyde (EMS) buffered in 0.1 M Na-P-buffer overnight, washed ×3 in 0.1 M buffer, post-fixed in 1% osmium tetroxide (Sigma-Aldrich) and dehydrated in ascending concentrations of ethanol followed by propylene oxide and infiltration in 30% and 50% Epon (Sigma-Aldrich).

At least three 0.9 μm toluidine blue stained semithin sections per localization were produced. Representative areas were trimmed and 90 nm lead citrate (Merck) and uranyl acetate (Merck) contrasted ultrathin sections were produced and subsequently viewed under Phillips CM10, operating with a Gatan Orius Sc1000 (832) digital camera, Gatan Microscopical Suite, Digital Micrograph, Version 230.540.

### 2.12. Preparation of PFCOC Emulsion

Perfluorooctylbromide (PFOB) was distilled twice and detoxified on an alumina column, followed by filtration on a 0.22 µm Millipore filter prior to use. Egg yolk phospholipids were used as the main emulsifier (4.5% weight/volume). The advanced PFOB nanoemulsion (90% PFOB weight/volume) was stabilized using the fluorocarbon/hydrocarbon diblock C_6_F_13_C_10_H_21_ (*F*6*H*10) (2.8% weight/volume). *FnHm* diblocks help to (i) control the emulsion droplets’ mean diameter, (ii) strongly stabilize the emulsion (no significant difference in droplet size was detected after one year at 25 °C), and (iii) provide well-tolerated emulsions that were demonstrated to be efficient [47,48,49,50]. The emulsion batches were prepared by high-pressure emulsification (Microfluidizer, under nitrogen) and heat-sterilized (121 °C, 103 kPa, 15 min). The droplet mean diameter was ~0.15 µm with a narrow size distribution (~25%) after heat sterilization. Osmolality and pH were adjusted at 304 mOsm and 7.1, respectively. Viscosity was 5 cP (extrapolated at zero shear rate) at 25 °C. The PFOB emulsion is stable for more than two years at room temperature. Compatibility with the Steen solution was verified. No effects on nanodroplet size and size distribution were observed over a period of 30 days in 10% and 30% dilutions in this medium.

### 2.13. Timing and Dosage of PFCOC

When we reached perfusion temperature of 37 °C, PFCOC nanoemulsion (0.1 g/kg of FBOB) was injected into the EVLP circuits in a final volume of 20 mL Steen solution via the pulmonary artery line.

## 3. Statistical Analysis

Statistical analyses were performed with PRISM 8 software (GraphPad Software, Inc., La Jolla, CA, USA). All results are expressed as mean ± standard deviation (SD). The Mann–Whitney test was utilized where data were non-continuous. Two-way analysis of variance (ANOVA) for repeated measures was utilized where such data contained a time component. In the transplantation groups, the immediate transplantation (Im-Tx) group was not compared with the other two groups. Differences are considered significant when the *p*-value is less than 0.05.

## 4. Results

### 4.1. EVLP Only Groups

#### Donor Characteristics

Donor body weight (control 29.80 ± 3.63 kg versus PFCOC 30.29 ± 4.07 kg), donor PaO_2_/FiO_2_ ratio (control 65.8 ± 3.7 kPa versus PFCOC 60.7 ± 5.6 kPa), donor lung compliance (control 23.22 ± 2.51 mL/mbar versus PFCOC 24.56 ± 4.64 mL/mbar) and cold ischemic time (control 24.19 ± 0.24 h versus PFCOC 24.07 ± 0.11 h) did not differ significantly between the two groups. Only perfusate consumption was significantly lower in the PFCOC group (control 538.3 ± 146.2 mL versus PFCOC 365 ± 97 mL) (*p* = 0.0455).

Lung physiology and metabolism were monitored during the 6 h EVLP period. Delta PO_2_ (control 36.48 ± 3.84 kPa versus PFCOC 44.13 ± 6.65 kPa) (*p* = 0.188), pulmonary vascular resistance (control 9.95 ± 1.91 Wood units versus PFCOC 9.42 ± 1.65 Wood units) (*p* = 0.7) and lung compliance (control 15.9 ± 1.65 mL/mbar versus PFCOC 15.29 ± 2.93 mL/mbar) (*p* = 0.8) were comparable in the two experimental groups.

Perfusate electrolytes, including potassium and sodium were gradually increased in both groups throughout the EVLP. However, potassium ion concentrations tended to be lower in the PFCOC group than in the control group (control 4.70 ± 0.25 mmol/L versus PFCOC 4.45 ± 0.13 mmol/L) (*p* = 0.18).

Perfusate glucose levels were constantly decreased in both groups (control 145.2 ± 17.66 mg/dL versus PFCOC 148.4 ± 17.49 mg/dL) (*p* = 0.52). Furthermore, lactate levels also increased gradually in both groups throughout the perfusion. However, lactate levels tended to decrease in the PFCOC group compared to the control group, without statistical significance (control 4.89 ± 2.53 mmol/L versus PFCOC 4.23 ± 2.26 mmol/L) (*p* = 0.12).

In bronchoalveolar lavage (BAL) cell count and differential count at the end of 6 h EVLP, we observed a trend towards a reduction in the total cell count (control 1152 ± 609 cells/uL versus PFCOC 657 ± 454 cells/uL, *p* = 0.18) and macrophages (control 80.75 ± 9.5% versus PFCOC 78.64 ± 9.5%, *p* = 0.55) but there were more neutrophils (control 2.5 ± 1.8% versus PFCOC 3.78 ± 2.9%, *p* = 0.41) and lymphocytes (control 10.33 ± 7.49% versus PFCOC 17.29 ± 8.45%, *p* = 0.21) in the PFCOC group.

ATP lung tissue levels tended to be higher in the PFCOC group compared to the control group. However, the difference was not statistically significant (control 844.9 ± 1694 nmol/mg protein versus PFCOC 1415 ± 1125 nmol/mg protein) (*p* = 0.12).

Myeloperoxidase (MPO) activity in lung tissue at the end of 6 h EVLP was lower in the PFCOC group (3.55 ± 2.4 mU/mL) compared to the control group (7.49 ± 2.4 mU/mL) without reaching statistical significance (*p* = 0.12).

Radiologic assessment score revealed significantly more lung infiltrates in the control group than in the PFCOC group at the end of EVLP (control 5.83 ± 2.22 versus PFCOC 3.85 ± 1.46) (*p* = 0.0495) (Figure 1).

### 4.2. Results for Transplantation Groups

Donor characteristics including donor body weight (control 29.80 ± 3.63 kg versus PFCOC 31.5 ± 4.65 kg), donor PaO_2_/FiO_2_ ratio (control 65.89 ± 3.41 kPa versus PFCOC 60 ± 6.68 kPa), donor lung compliance (control 23.66 ± 2.53 mL/mbar versus PFCOC 24.85 ± 4.75 mL/mbar), cold ischemic time (control 24.13 ± 0.21 h versus PFCOC 24.1 ± 0.14 h) and perfusate consumption did not differ significantly between the groups (control 536 ± 163.3 mL versus PFCOC 351.3 ± 104.9 mL) (*p* = 0.127).

Lung physiology and metabolism were monitored during the 6 h EVLP period.

Delta PO_2_ (control 35.29 ± 3.38 kPa versus PFCOC 48.49 ± 7.63 kPa) (*p* = 0.1), lung dynamic compliance (control 16.37 ± 1.82 mL/mbar versus PFCOC 15.25 ± 4.21 mL/mbar) (*p* = 0.7), and pulmonary vascular resistance (control 9.93 ± 1.73 Wood units versus PFCOC 9.60 ± 1.91 Wood units) (*p* = 0.8) were comparable between groups (Figure 2).

Perfusate electrolytes, including potassium and sodium increased gradually in both groups throughout the EVLP. However, potassium ion concentrations tended to be lower in the PFCOC group than in the control group (control 4.70 ± 0.25 mmol/L versus PFCOC 4.32 ± 0.08 mmol/L) (*p* = 0.17).

Perfusate glucose levels were constantly decreased in both groups (control 144.5 ± 18.12 mg/dL versus PFCOC 149.7 ± 16.21 mg/dL) (*p* = 0.3). Furthermore, lactate levels also increased gradually in both groups throughout the perfusions. However, lactate levels were significantly lower in the PFCOC group (control 4.91 ± 2.55 mmol/L versus PFCOC 3.73 ± 2.07 mmol/L) (*p* = 0.0249) (Figure 3).

In BAL, we observed a trend towards fewer total cells (control 1152 ± 681 cells/uL versus PFCOC 410 ± 225 cells/uL, *p* = 0.11) and macrophages (control 81.3 ± 10.24% versus PFCOC 76.75 ± 9.77%, *p* = 0.46) but there were more neutrophils (control 2.5 ± 2% versus PFCOC 4.62 ± 3.6%, *p* = 0.25) and lymphocytes (control 9.4 ± 7.98% versus PFCOC 18.13 ± 6.68%, *p* = 0.16) in the PFCOC group.

Perfusate cytokine levels are shown in Table 1.

Hemodynamic and physiologic parameters during the 4 h of reperfusion are shown in Figure 4. The control group had worse mean arterial pressure during the reperfusion period compared to the treated group (control 52.83 ± 7.68 mmHg versus PFCOC 61.54 ± 11.88 mmHg) (*p* = 0.035). The systemic arterial gas exchange (control 61.03 ± 4.57 kPa versus PFCOC 65.89 ± 9.08 kPa), allograft pulmonary graft venous oxygen concentration (control 55.06 ± 1.59 kPa versus PFCOC 73.72 ± 4.45 kPa), and dynamic compliance (control 19.54 ± 3.05 mL/mbar versus PFCOC 22.89 ± 2.04 mL/mbar) were comparable between the control and treated group.

At the end of the transplantation, following clamping of the right pulmonary artery, gas exchange was significantly higher in PFCOC group compared to the control (control 36.96 ± 16.95 kPa versus PFCOC 68.91 ± 3.59 kPa) (*p* = 0.0159) (Figure 5). When we clamped the right main bronchus, the compliance was similar in the two groups (control 7.74 ± 3.24 mL/mbar versus PFCOC 10.75 ± 3.6 mL/mbar) (*p* = 0.2) (Figure 5).

Table 2 shows the plasma cytokine levels at the end of reperfusion. Plasma IL-8 and IL-12 levels were significantly lower in the PFCOC group compared to the control (*p* = 0.01, and *p* = 0.03, respectively).

BAL IL-1α (control 20.6 ± 13.1 pg/mL versus PFCOC 9.1 ± 16.6 pg/mL, *p* = 0.11), IL-1β (control 583.5 ± 563.1 pg/mL versus PFCOC 104.8 ± 91.23 pg/mL, *p* = 0.06), IL-1Ra (control 3057 ± 2896 pg/mL versus PFCOC 439.4 ± 289.8 pg/mL, *p* = 0.11), and IL-12 (control 159.1 ± 107 pg/mL versus PFCOC 102.6 ± 83.21 pg/mL, *p* = 0.7) were markedly reduced in the PFCOC group compared to the control (Table 3).

ATP lung tissue levels at the end of the transplantation (after 4 h of reperfusion) were numerically higher in the PFCOC group compared to the control group (control 716 ± 823.7 nmol/mg protein versus PFCOC 1300 ± 1344 nmol/mg protein, *p* = 0.5).

MPO levels in lung tissue after 4 h reperfusion in the transplant groups tended to be lower in the PFCOC group (4.89 ± 2.5 mU/mL) compared to the control group (9.03 ± 2.8 mU/mL, *p* = 0.19). In the immediate transplant group MPO lung tissue levels were 0.01 ± 0.01 mU/mL.

The level of flavin mononucleotide (FMN) in the EVLP perfusate was significantly higher in the control group compared to PFCOC group: time 0 (baseline): 5489 ± 607 versus 5327 ± 458; EVLP 1 h: 5309 ± 247 versus 4881 ± 412; EVLP 2 h: 5805 ± 326 versus 5282 ± 570; and EVLP 3 h: 6227 ± 221 versus 5827 ± 351 (*p* = 0.0089), respectively (Figure 6).

### 4.3. Histology of the Transplantation Groups

The control group exhibited a mild early inflammatory reaction with activated endothelium and intravascular neutrophils. There was definite evidence of neutrophil emigration, which showed intense alveolar damage (Table 4, Figure 7).

With regard to the histology of the PFCOC group, light microscopy of alveolar septa showed only a few apoptotic/karyorrhectic pneumocytes. The number of macrophages in the septa was not significant (Figure 8A). Blood vessels showed a small number of neutrophils in the lumen and activated endothelial cells (Figure 8B). There were also a few emigrated neutrophils. In TEM sections, the normal pig lung showed no evidence of inflammatory changes (Figure 9A,B). Endothelial cells and to a lesser extent type I pneumocytes exhibited pinocytic vesicles. TEM images of the treated (PFCOC) group showed very good coverage of vital alveolar lining cells, prominent type II pneumocytes, moderate numbers of pulmonary intravascular macrophages, and a low number of intracapillary RBCs and neutrophils (Figure 10A,B). Vesiculation of capillary endothelial cells and type I pneumocytes was obvious. We interpreted this as prominent gas exchange (Figure 10 C,D).

## 5. Discussion and Conclusions

▪In this experimental study, we tested the protective effect of PFCOC emulsion given during EVLP. We observed that metabolic markers of donor quality such as lactate production, glucose consumption and preservation of mitochondrial function were better in the PFCOC group. Following transplantation, the PFCOC group demonstrated better graft function and TEM showed better tissue preservation and cellular viability.▪As the number of available organs is less than the number of the patients on the waiting list, the transplant community continues to search for alternatives to increase the number of available lung donors. These include extended criteria donors, DCD or utilizing EVLP for assessment of questionable lungs prior to implantation. A successful transplantation begins with the optimal preservation of the organ. In lung transplantation, the current standard is the infusion of cold organ preservation solution followed by cold static storage. In cold static storage, the metabolism is decreased, however, energy consumption continues and the hypoxic state activates anaerobic pathways, which results in cell death [51].▪Perfluorocarbons (PFCs) are a class of chemicals that are essentially composed of carbon and fluorine atoms [36,38,39,52]. The fluorine atoms contribute to their high chemical and thermal stability [52]. Two essential features of the PFC are their unique gas-dissolving capacity and their exceptional chemical and biological “inertness” [52]. Their gas-dissolving capacity is a consequence of the weakness of the intermolecular forces that prevail in liquid fluorocarbons, which facilitate the formation of “holes” that can accommodate gas molecules within the liquid. On the other hand, their inertness reflects the strength of the intramolecular chemical bonds [36,38,39,52].

PFC-based emulsions have the capacity to dissolve large concentrations of oxygen, they can be produced with small particle sizes, and they are low-viscosity suspensions [53,54,55,56]. The capacity of PFC liquids to support the life of rats submerged in PFC liquids was shown by Clark and Gollan in 1966 [57]. Oxygen solubility is typically 40–60 mL of gas dissolved per 100 mL of PFCs whereas that for carbon dioxide can be up to three times higher [58]. PFCs are hydrophobic and an emulsified form is necessary for intravenous use. It is currently feasible to produce stable PFC emulsions with particles of median diameter < 0.2 μm [59]. Most PFC emulsions have average diameters of 0.2 μm compared with 5–7 μm for erythrocytes, and particles of this size can access spaces between red blood cells and vessel walls and perfuse even constricted micro vessels [60].

The anti-inflammatory effects of PFCs either by direct modulation of inflammatory cell function or attenuation of the production of inflammatory mediators have been reported [40,41,42,43,44]. In our experiments, we also observed an anti-inflammatory effect. At the end of 6 h EVLP, perfusate pro-inflammatory cytokines such as IL-1α, IL-1β, TNF-α and IL-1Ra were lower in the PCOC group. At the end of transplantation, plasma proinflammatory cytokines such as IL-8 and IL-12 were significantly lower the PFCOC group. In addition, although not reaching to statistical significance TNF-α, IL-6, IL-1α, IL_1Ra and IL-1β were lower in the PFCOC group.

Chu et al. used an isolated rat lung perfusion model to demonstrate the beneficial effects of systemic PFC administration on ischemia reperfusion injury ^40^. This effect was explained by the inhibition of NF-kB activity, inflammatory reactions and neutrophil infiltration into the lung tissue [40]. Systemic administration of PFC reduced phorbol myristate acetate-induced acute lung injury [61]. In isolated and perfused rat lung model, Fluorinert (FC-77), a perfluorocarbon compound, attenuated the lung injury in a dose-dependent manner. The mechanism is thought to be due to decreasing the inflammatory response activated by neutrophils [61]. Schroeder et al. demonstrated that PFCs increase spinal cord oxygen delivery which offers promise for decreasing ischemic injury [62]. The small size of PFC emulsion particles gives them the ability to deliver oxygen to tissue beds that are unreachable for erythrocytes [63]. In rat model of traumatic brain injury, Mullah et al. demonstrated an increase in brain tissue oxygen tension in the group treated with PFC compared to a control group [63]. PFCs reduced infarct size in mice and rabbit models of myocardial ischemia [64,65,66]. PFC improved tissue oxygenation in canine models [67,68] and reversed physiologic transfusion triggers in surgical patients [69]. A PFOB nanoemulsion improved myocardial oxygenation and cardiac function in the isolated working rabbit heart [70]. Administration of a PFC emulsion reduced infarct size and improved functional recovery after acute ischemic stroke in the rat [71]. Resuscitation of mice from hemorrhagic shock with a PFOB emulsion was comparable to that achieved with whole blood [72]. Acute lung injury in rats was relieved upon intravenous administration of a PFOB nanoemulsion [73]. A PFOB nanoemulsion improved the survival of rats after liver transplantation [74].

In a phase 3 study, PFC decreased allogeneic erythrocyte transfusions in major non-cardiac surgical procedures associated with significant blood loss [69,75]. In experimental hemorrhagic shock, perflubron emulsion restored hepatic microcirculatory flow and oxygenation while supporting blood pressure [76,77].

As PFCs are not metabolized, but excreted unchanged via the lungs, their potential for cytotoxicity is thought to be limited [78]. There is no antigenicity. However, because PFCs are taken up avidly by the reticuloendothelial system, they increase liver enzymes and can result in hepatosplenomegaly when the droplet size is too large [36,79]. Because of the extensive uptake in the reticuloendothelial system and impairment of neutrophil function, they may interfere with host-defense mechanisms. Monocyte and macrophage activation may lead to release of prostaglandins, endoperoxides, and cytokines, which probably accounts for symptoms such as flushing, backache, fever, chills, headaches, and nausea observed in clinical trials [78]. Platelet count decreases by as much as 40% because of increased platelet clearance from PFC-induced modification of platelet surfaces [80]. Perflubron emulsion attenuated simulated extracorporeal circulation-induced neutrophil activation as evidenced by the reduction in adhesion to nylon fiber columns [81].

In our study, the control group exhibited mild early inflammatory reaction (activated endothelium and intravascular neutrophils) and evidence of neutrophil emigration (Table 4). Lower MPO levels, a marker of neutrophil infiltration, and low plasma IL-8 levels also supported reduced neutrophil activation in the PFCOC group. During 6 h of EVLP and at the end of transplantation, we had comparable IL-6 levels in the perfusate and in the plasma. This could be the effect of PFCOC in the perfusate and its somewhat protective effect on the endothelial cells. However, only in the BAL at the end of transplantation IL-6 levels were higher in the treated group compared to control. We may speculate that alveolar cells might react differently to secrete IL-6 in response to inflammation.

Intravenous infusion of perfluorocarbon, did not cause additional vasoconstriction in cerebral pial arterioles or increase systemic blood pressure compared with saline and colloid in healthy anesthetized rat cranial window method [82]. This is an important observation. In our study, pulmonary vascular resistance during EVLP was comparable between the two groups, which supports the observation made by Abutarboush et al. [82].

Use of PFCOC emulsion prevented the adhesion of established pancreatic beta cell lines and the oxygen-carrying capacity of PFCOC emulsion may also help to prevent the period of hypoxia exposure, which is known to contribute to the failure of post-transplant survival [83]. Glucose is the major source of energy for the cells for special tasks such as protein synthesis, nucleic acid replication and other intracellular processes [84]. Glycolysis is defined as the anaerobic process of ATP production [84]. Lactate is the end product of the glycolysis. Lactate inhibits glycolysis and can affect the lung viability by reducing glucose availability [85]. In a pig model, high glucose consumption showed impaired lung function and edema [86]. Koike et al. showed that lactate production and washout is a continuous process, and lactate/pyruvate (L/P) and glucose levels could be indicators for poor prognosis [87]. However, this group was not able to find any correlation between L/P ratio and lung transplant outcome in their clinical lung transplant program [88]. Global tissue hypoxia is the main cause of excess lactate production and mostly due to hypoperfusion [84].

In accordance to the above-mentioned experimental and clinical observations in our study, perfusate glucose levels decreased constantly in both groups, but the control group showed a trend toward more glucose than the PFCOC group. In addition, lactate levels also increased gradually in both groups throughout the perfusion. However, lactate levels were lower in the PFCOC group compared to the control group.

Flavin mononucleotide (FMN) is a small molecule bound to the mitochondrial complex I [89]. The loss of FMN results in a decline in complex I activity, and thus, mitochondrial dysfunction [90]. FMN release is reported to be a marker for ischemia reperfusion injury in hypothermic or normothermic oxygenated perfusion solution [46,91]. FMN levels in the acellular perfusate after 30 min of hypothermic perfusion was predictive of 90-day graft loss [91]. In our PFCOC group, the FMN release was lower compared to the control, which indicates less mitochondrial damage. In a clinical pilot study, lungs did not release much FMN during normothermic machine perfusion compared to liver and kidneys [46]. It seems that in the future, FMN could be a real-time marker for graft acceptance during machine perfusion.

In our study, following lung transplantation, the lungs principally exhibited the same morphological features. There was some degeneration and cell death, and we observed the presence of intracapillary macrophages (pulmonary resident macrophages, i.e., pulmonary intravascular macrophages) and macrophages that form small clusters in the interstitial space.

The immediate transplant group had a higher number of macrophages in the alveolar septa compared to the control and treated groups, which might be a consequence of EVLP flushing. The macrophages were shown to proliferate. These features are also seen in EVLP groups only. Transplantation is associated with an acute inflammatory reaction, represented by endothelial cell activation, neutrophil rolling and emigration. The extent of this inflammatory response appears to be lowest in the immediate transplantation group. However, there was a tendency for a less intense inflammatory reaction in PFCOC group.

TEM showed another interesting feature, that is, there was distinct vesiculation of capillary endothelial cells and to a lesser extent, vesiculation in type I pneumocytes in the PFCOC group. We interpreted this finding as evidence of trans-membranous transport or gas exchange. It appears to be attributable to PFCOC treatment, since otherwise endothelial cell vesiculation was only seen in normal lung and in the treated group.

In conclusion, perfluorocarbon-based oxygen carrier (PFCOC) application is safe during EVLP in lungs preserved 24 h at 4 °C. Although this strategy did not significantly affect the EVLP physiology, metabolic markers of donor quality such as lactate production, glucose consumption and preservation of mitochondrial function were better in the treatment group. Inflammation was lower in the treatment group. Moreover, perfusate enrichment improves electrolyte imbalance to a certain degree during EVLP. Following transplantation, PFCOC provided better graft function and TEM showed better tissue preservation and cellular viability compared to controls.

## Figures and Tables

**Figure 1 cells-09-02501-f001:**
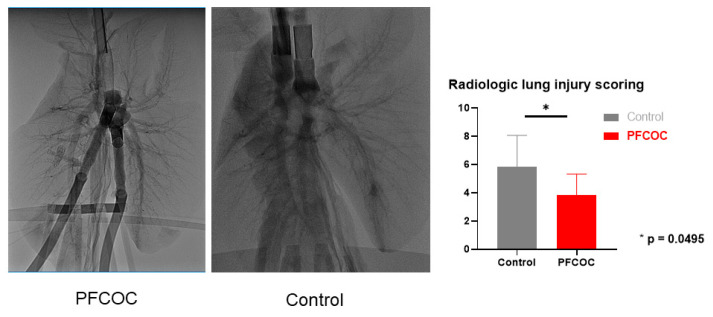
Radiologic assessment score. Control group showed more infiltrates than PFCOC group at the end of 6 h ex vivo lung perfusion (EVLP). The Mann–Whitney test was utilized. Control: control group, PFCOC: perfluorocarbon-based oxygen carrier group.

**Figure 2 cells-09-02501-f002:**
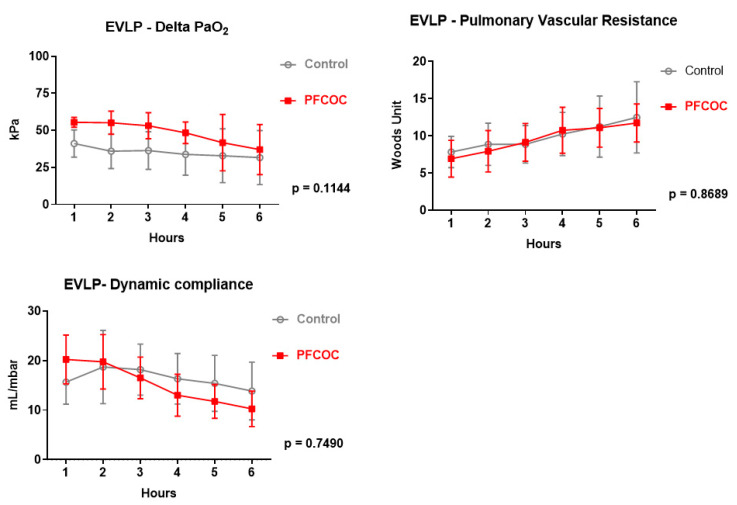
Lung physiology and metabolism were monitored during the 6 h EVLP period. Two-way analysis of variance (ANOVA) was used for repeated measures. Control: control group, PFCOC: perfluorocarbon-based oxygen carrier group.

**Figure 3 cells-09-02501-f003:**
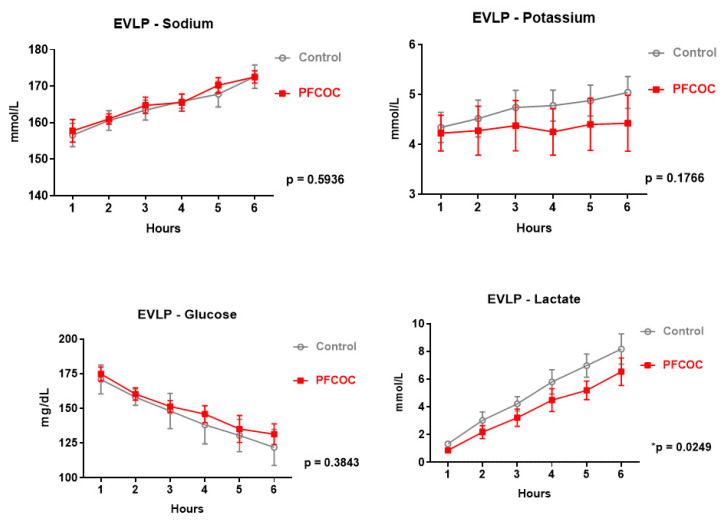
Perfusate electrolytes, glucose and lactate levels in both groups throughout the EVLP. Two-way analysis of variance (ANOVA) for repeated measures. Control: control group, PFCOC: perfluorocarbon-based oxygen carrier group.

**Figure 4 cells-09-02501-f004:**
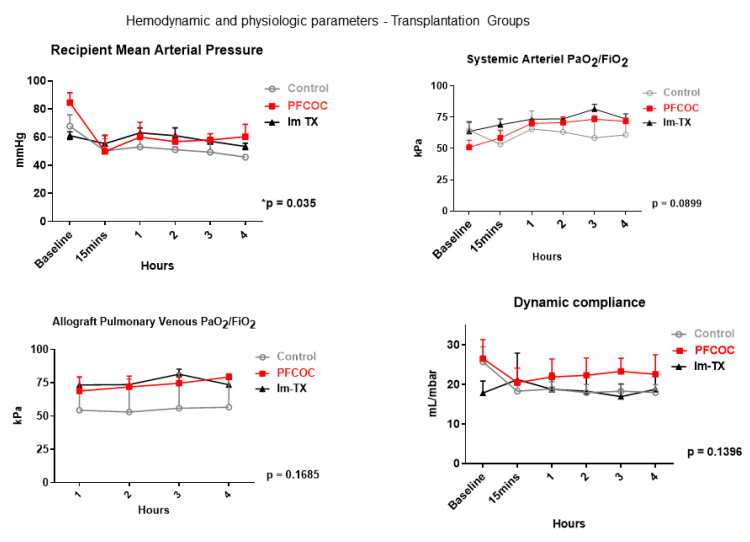
Hemodynamic and physiologic parameters during the 4 h of reperfusion. Im-Tx: Immediate transplantation group. Two-way analysis of variance (ANOVA) for repeated measures. Control: control group, PFCOC: perfluorocarbon-based oxygen carrier group.

**Figure 5 cells-09-02501-f005:**
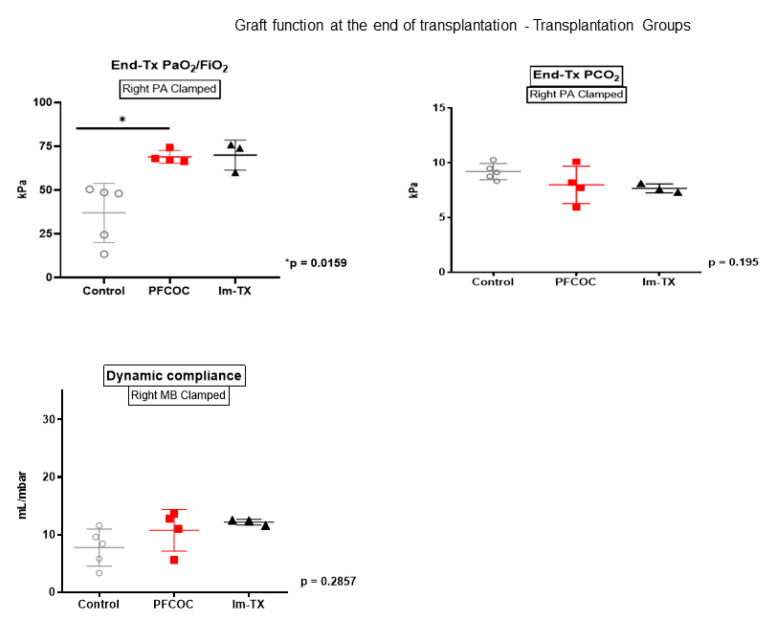
Graft function at the end of the 4 h reperfusion period. The Mann–Whitney test was used. Control: control group, PFCOC: perfluorocarbon-based oxygen carrier group. Im-Tx: Immediate transplantation group. The Im-Tx group was not included in the statistical analysis.

**Figure 6 cells-09-02501-f006:**
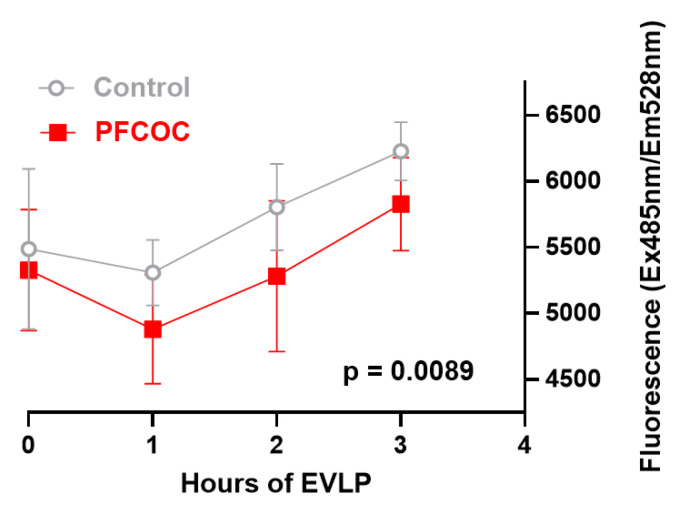
The flavin mononucleotide fluorescence readings of the perfusate taken hourly during EVLP is significantly higher in the control group compared to PFCOC group. Two-way analysis of variance (ANOVA) for repeated measures. Control: control group, PFCOC: perfluorocarbon-based oxygen carrier group.

**Figure 7 cells-09-02501-f007:**
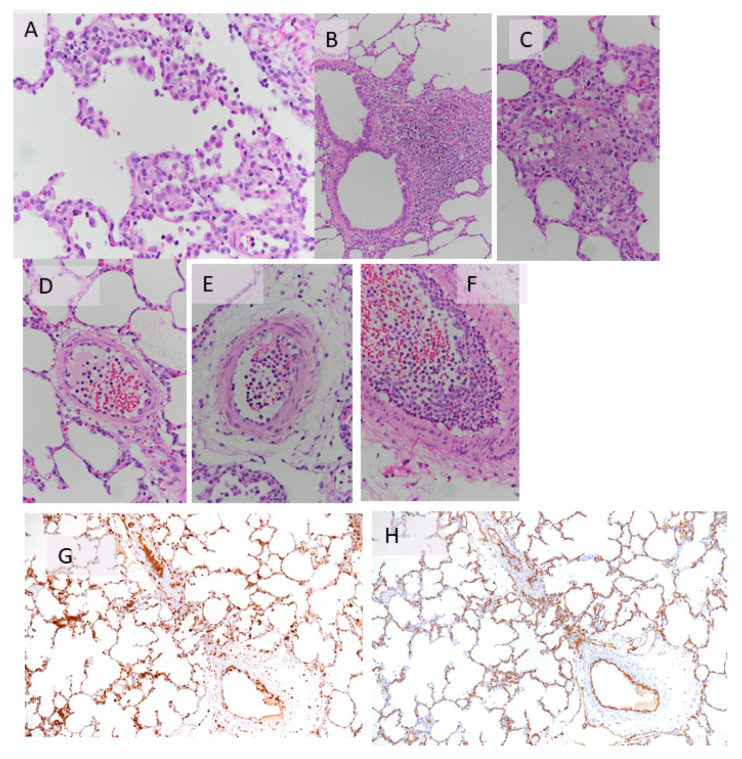
(**A**–**H**) Histology of the control group at the end of 4 h reperfusion showed that there was moderate to severe multifocal alveolar damage with desquamated alveolar macrophages/type II pneumocytes in the alveolar lumen, also admixed with occasional neutrophils ((**A**–**C**): HE Staining). Occasional apoptotic/karyorrhectic cells are observed, especially interstitial macrophages ((**D**–**F**): HE staining). Alveolar septa are occasionally expanded by intracapillary leukocytes (**G**); ionized calcium binding adaptor molecule 1 (Iba-1) staining, (**H**): CD31 staining).

**Figure 8 cells-09-02501-f008:**
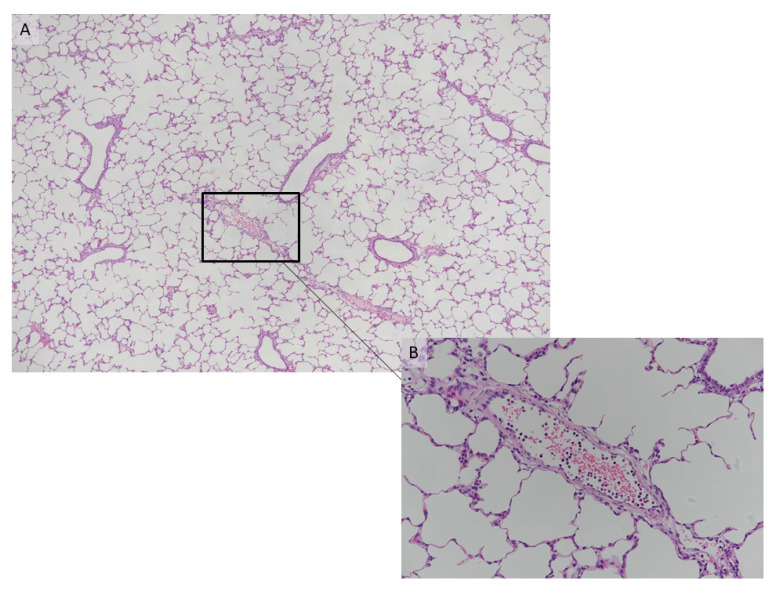
(**A**,**B**) Light microscopy of alveolar septa showed only a few apoptotic/karyorrhectic pneumocytes. The number of macrophages in the septa was not significant (**A**). Blood vessels showed a small number of neutrophils in the lumen and activated endothelial cells (**B**) (HE staining).

**Figure 9 cells-09-02501-f009:**
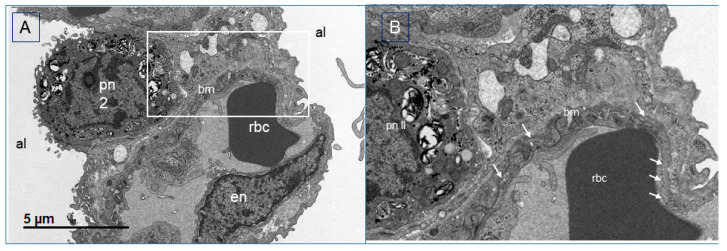
(**A**,**B**) TEM of normal pig lung. Normal pig lung represents the lung tissue that was taken during the pneumonectomy of the recipient left lung. Pulmonary intravascular macrophages and intracapillary RBCs are present, the pneumocyte coverage appears adequate. There is no evidence of degenerative or other, i.e., inflammatory changes. Endothelial cells and to a lesser extent type I pneumocytes exhibit pinocytic vesicles. This sample is an example of normal structure with unaltered cells in the pig lung. There is evidence of gas exchange (more on the endothelial side). (**A**) Mildly dilated capillary with single rbc, and (**B**) close-up view of the white rectangle in (**A**) showing the endothelial pinocytic vesicles (arrows). Pn I: Pneumocyte type I, pn II: Pneumocyte type II, bm: basal membrane, rbc. Red blood cell, en: endothelial.

**Figure 10 cells-09-02501-f010:**
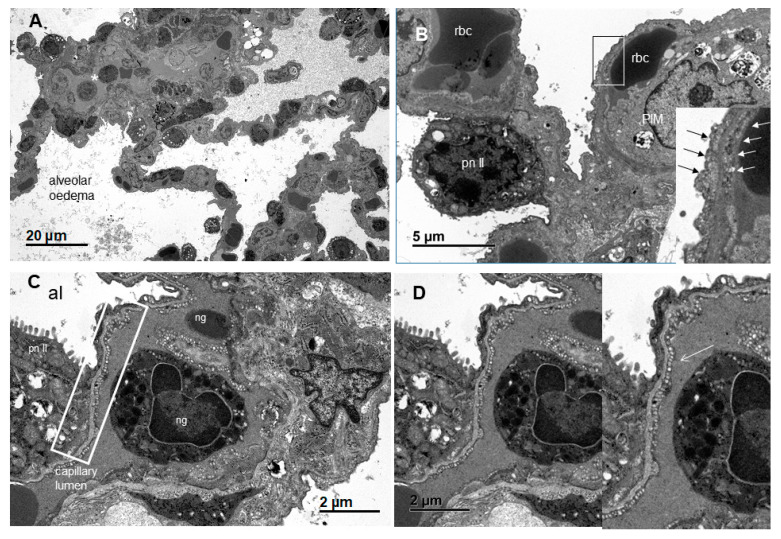
TEM images of the treated (PFCOC) group. (**A**,**B**) Very good coverage of vital alveolar lining cells, prominent type II pneumocytes, moderate numbers of pulmonary intravascular macrophages (PIMs), and a low number of intracapillary RBCs and neutrophils were observed. There were frequent endothelial pinocytic vesicles that indicated activation and intense transmembranous transport. Vesiculation was also seen in type I pneumocytes. Vesiculation of capillary endothelial cells and type I pneumocytes was obvious. This is interpreted as prominent gas exchange (*): intracapillary leukocyte. (**C**,**D**) There was distinct vesiculation of capillary endothelial cells, and to a lesser extent, vesiculation was also seen in type I pneumocytes. This was evidence of transmembranous transport/gas exchange. It appeared to be attributable to PFOB treatment, since otherwise endothelial cell vesiculation was only seen in normal lung as shown in Figure 9A,B.

**Table 1 cells-09-02501-t001:** Perfusate cytokine levels at the end of 6 h EVLP.

Cytokine	Control	PFCOC	*p* Value
TNFα	335.1 ± 335.2	161.6 ± 76.9	0.6667
IL1-α	68 ± 46.8	28 ± 20.9	0.2857
IL1-β	982 ± 777.1	599.6 ± 345.1	0.5556
IL1-Ra	944.7 ± 384.3	703 ± 461.3	0.7302
IL-2	1224.9 ± 922.7	1439 ± 466.4	0.7302
IL-4	6108.8 ± 5479.9	6657.6 ± 2611	0.9999
IL-6	3612.3 ± 1786.4	3680.2 ± 910.4	0.9999
IL-8	33901.1 ± 10995.8	38632.6 ± 15126	0.7302
IL-10	871.4 ± 796.4	935.6 ± 376.8	0.5556
IL-12	438.6 ± 130.9	346.4 ± 72.8	0.2857
IL-18	10845.1 ± 3782.1	10142.8 ± 4613	0.7302

Values are given as mean ± standard deviation.

**Table 2 cells-09-02501-t002:** Plasma cytokine levels at the end of 4 h reperfusion after transplantation.

	Control	PFCOC	*p* Value
IFNγ	1605.3 ± 264.4	1602.6 ± 509.7	0.9048
TNFα	72.4 ± 53.1	23.8 ± 9.9	0.44
IL1-β	355.8 ± 244	261.9 ± 136	0.7302
IL1-Ra	12.1 ± 20.8	8.7 ± 10.9	0.9762
IL-4	12.1 ± 20.8	221.5 ± 284.4	0.9048
IL-6	331.3 ± 55	334.2 ± 143	0.7302
IL-8	477.2 ± 892	3.3 ± 3.9	0.0159
IL-10	226.5 ± 283.2	160.3 ± 137.1	0.7302
IL-12	1116.3 ± 116.5	916.3 ± 105.9	0.0317
IL-18	524 ± 498.9	572 ± 317.1	0.7302

Values are given as mean ± standard deviation.

**Table 3 cells-09-02501-t003:** Bronchoalveolar lavage (BAL) cytokine levels at the end of 4 h reperfusion after transplantation.

	Control	PFCOC	*p* Value
IL1-α	20.6 ± 10	9.1 ± 16.6	0.1111
IL1-β	583.5 ± 563.1	104.8 ± 91.2	0.0653
IL1-Ra	3057 ± 2896	439.4 ± 289.8	0.1111
IL-2	29.7 ± 26.7	529.6 ± 779.4	0.0635
IL-4	73.5 ± 92.9	2269 ± 1909	0.7302
IL-6	601.5 ± 123	931.2 ± 194.7	0.0317
IL-8	3064 ± 2160	3118 ± 3526	0.5556
IL-10	24 ± 25.3	248.6 ± 419.8	0.5556
IL-12	159.1 ± 107.1	102.6 ± 83.2	0.7302
IL-18	272.2 ± 96	2282 ± 2687	0.1905

Values are given as mean ± standard deviation.

**Table 4 cells-09-02501-t004:** Histologic score for inflammatory process observed after lung transplantation.

Group	NL in Vessels	NL Emigration	NL in Alveoli	Total Score	Average Score
CONT	1	1	1	3	
CONT	1.5	1	1	3.5	
CONT	2	2	3	7	5.2
CONT	1	1	1.5	3.5	
CONT	3	3	3	9	
PFCOC	1	1	0.5	2.5	
PFCOC	2	1.5	0	3.5	2.8
PFCOC	1	1.5	0	2.5	
PFCOC	1.5	1.5	0	3

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
