# Peer review of "Functional, Metabolic and Morphologic Results of Ex Vivo Donor Lung Perfusion with a Perfluorocarbon-Based Oxygen Carrier Nanoemulsion in a Large Animal Transplantation Model"

_cells, 2020, doi:10.3390/cells9112501_

Round 1

Reviewer 1 Report

An excellent contribution to the EVLP community. The manuscript should definitely be published in its present form. 

Author Response

We thank the reviewer for this knd comment.

Reviewer 2 Report

The present study investigated effects exerted by perfluorocarbon administration during porcine EVLP. Graft function was assessed both during EVLP procedure and following lung transplantation by evaluating pulmonary gas exchange, lung mechanics and vascular pressures. Perfusate gas-analysis, histology, and transmission electron microscopy (TEM) were likewise performed. Perfusate, plasma, and BAL inflammatory mediator concentration was measured using an immunofluorescent assay. Finally, perfusate flavin mononucleotide (FMN) was investigated.

Major

  1. Why is numerosity not consistent across experimental groups? Please, provide information on sample size calculation
  2. Lactate concentration measured during EVLP of the “transplantation groups” is significantly different between PFCOC and control groups. Why this difference was not revealed in “EVLP only groups”? Methodological timeframes appear to be similar: 24 h cold storage + 6 h EVLP. Please, provide an explanation for this inconsistency
  3. There is a statistically significant increase in BAL IL-6 in the PFCOC compared to the control group; please, discuss this result

Minor

  • Give details on MPO activity assessment indicating if fresh lung biopsies were used
  • ATP content is expressed as nmol/mg protein; describe how protein concentration was evaluated in tissue samples and indicate if it was measured in the same lung biopsy used to assess ATP
  • Pag 6: replace MPO levels with MPO activity
  • Format correctly titles at pag 5 and 6
  • Add group numerosity and test used to perform statistical analysis in the figure captions
  • Please explain the abbreviation Im-Tx in the “Study groups” paragraph.
  • P value for plasma TNF-alpha reported in the Table 2 pag 10 is different from the value reported in the text
  • Figure 9. Explain “normal pig”
  • Figure 10, Panel A and B: indicate which samples are analyzed in these pictures
  • Format correctly the paragraph “Comment”

Author Response

Answers to the reviewer #2:

Major

  1. Why is numerosity not consistent across experimental groups? Please, provide information on sample size calculation

For EVLP groups we designed the experiments as randomized six animals per group. In order to be more precise we added one more experiment in the PFCOC group.

Transplantation group experiments: We designed them as five experiments per group. As we saw the benefit in the treated group, we omitted the last experiment in the treated group.

Immediate transplantation group (Im-Tx): This group is planned to have the golden standard in transplantation.  This means we retrieved the lung and transplanted it after 4 hours of cold storage. Therefore, we performed only three experiments in this group. We did not include this group in the statistical evaluation.

This is added to page 3 in the methods section.

  1. Lactate concentration measured during EVLP of the “transplantation groups” is significantly different between PFCOC and control groups. Why this difference was not revealed in “EVLP only groups”? Methodological timeframes appear to be similar: 24 h cold storage + 6 h EVLP. Please, provide an explanation for this inconsistency

We thank the reviewer for this comment. Trend in the EVLP Groups was also better in the PFCOC group without reaching significance. In the transplantation groups, we observed the same trend, lower lactate in the treated group, reaching to significance. It is difficult to comment on this; if we made more experiments in the transplantation groups, we might have the same result as we had in the EVLP Only Groups. Main conclusion is, in these experimental setting lactate levels are lower in the treated (PFCOC) group.

This is already mentioned in the discussion at the end of page 16 and beginning of page 17.

  1. There is a statistically significant increase in BAL IL-6 in the PFCOC compared to the control group; please, discuss this result.

During 6 hours of EVLP and at the end of transplantation, we have comparable IL-6 levels, in the perfusate and in the plasma. This could be the effect of PFCOC in the perfusate and somewhat protective effect on the endothelial cells. However, only in the BAL at the end of transplantation IL-6 levels were higher in the treated group compared to control. We may speculate that, alveolar cells might react different to secrete IL-6 in response to inflammation.

This paragraph is added to the discussion section. Page 16.

Minor

  • Give details on MPO activity assessment indicating if fresh lung biopsies were used

Lung tissues were collected and flash frozen in liquid nitrogen and then stored at -80°C. Powdered frozen lung tissue (25 mg) were homogenized in 0.5 ml of 0.5% trichloroacetic acid and centrifuged at 8000 rpm for 2 min at 4°C. The supernatant were isolated and 10X concentrated Tris-acetate buffer was used to neutralize pH to 7.4 with a 10 microL of 0.002% xylenol blue used as pH indicator.

In page 4 we added this information.

  • ATP content is expressed as nmol/mg protein; describe how protein concentration was evaluated in tissue samples and indicate if it was measured in the same lung biopsy used to assess ATP

A microBCA protein assay kit was used with bovine serum albumin as standard to determine protein concentration in the supernatant of each sample tested according to the manufacturer’s instructions (Thermo Scientific, Rockford, Il, USA).

We measured in the same lung biopsy samples.

In Page 4 we added this information

  • Page 6: replace MPO levels with MPO activity

MPO levels is replaced as MPO “activity” in the text. Page 6.

  • Format correctly titles at pages 5 and 6

Thank you for this comment. Format is done by the editorial office automatically when they convert out text to their style.

  • Add group numerosity and test used to perform statistical analysis in the figure captions

They are added to the figures as proposed by the reviewer

  • Please explain the abbreviation Im-Tx in the “Study groups” paragraph.

Im-Tx: Immediate Transplantation. This is added to the text where indicated.

  • P value for plasma TNF-alpha reported in the Table 2 page 10 is different from the value reported in the text

This comment is correct. I corrected it in the text. I deleted the TNF-alpha

  • Figure 9. Explain “normal pig”

Normal pig lung represents the lung tissue that was taken during the pneumonectomy of the recipient left lung.  Page 14.

  • Figure 10, Panel A and B: indicate which samples are analyzed in these pictures

Those are the images from the same animal and treated group. We corrected it on the text. Page 15.

  • Format correctly the paragraph “Comment”

We tried to correct. As we mentioned before, format is done by the editorial office automatically when they convert out text to their style.